# Capsid-like particles decorated with the SARS-CoV-2 receptor-binding domain elicit strong virus neutralization activity

Cyrielle Fougeroux [iD] et al.[#]

The rapid development of a SARS-CoV-2 vaccine is a global priority. Here, we develop two capsid-like particle (CLP)-based vaccines displaying the receptor-binding domain (RBD) of the SARS-CoV-2 spike protein. RBD antigens are displayed on AP205 CLPs through a split-protein Tag/Catcher, ensuring unidirectional and high-density display of RBD. Both soluble recombinant RBD and RBD displayed on CLPs bind the ACE2 receptor with nanomolar affinity. Mice are vaccinated with soluble RBD or CLP-displayed RBD, formulated in Squalene-Water-Emulsion. The RBD-CLP vaccines induce higher levels of serum anti-spike antibodies than the soluble RBD vaccines. Remarkably, one injection with our lead RBD-CLP vaccine in mice elicits virus neutralization antibody titers comparable to those found in patients that had recovered from COVID-19. Following booster vaccinations, the virus neutralization titers exceed those measured after natural infection, at serum dilutions above 1:10,000. Thus, the RBD-CLP vaccine is a highly promising candidate for preventing COVID-19.

---

[#]A list of authors and their affiliations appears at the end of the paper.

Starting in December 2019, the severe acute respiratory syndrome corona virus 2 (SARS-CoV-2) outbreak rapidly spread, and by March 2020, the World Health Organization (WHO) declared a public health emergency of international concern[1]. SARS-CoV-2 belongs to the subfamily of *Coronavirinae* comprising at least seven members known to infect humans, including the highly pathogenic strains, SARS-CoV and Middle East respiratory syndrome corona virus (MERS-CoV)[2]. The symptoms of the disease (COVID-19) range from mild flu-like symptoms, including cough and fever, to life threatening complications. Both SARS-CoV and SARS-CoV-2 use highly glycosylated homotrimeric spike proteins to engage angiotensin-converting enzyme 2 (ACE2) on host cells to initiate cell entry[3–5]. The SARS-CoV spike proteins are known targets of protective immunity, eliciting both neutralizing antibodies and T cell responses upon natural infection[6]. Consequently, the spike protein is a primary target for SARS-CoV-2 vaccine development, with emphasis on the receptor-binding domain (RBD), which appears to be the target for most neutralizing antibodies[7–12]. The urgent need of an effective SARS-CoV-2 vaccine, to contain the worldwide pandemic and prevent new viral outbreaks, has led to a global effort involving a wide range of vaccine technologies. These include genetic-based (mRNA and DNA) principles[13,14], replicating/non-replicating viral vectors (measles[15], adenovirus[16,17], baculovirus), recombinant proteins or peptides[18], virus-like particles (VLPs)/nanoparticles or inactivated and live-attenuated viral vaccines[19–21]. In fact, more than 120 SARS-CoV-2 vaccine candidates are currently registered by WHO, of which 41 are currently undergoing clinical testing[22]. We have developed a SARS-CoV-2 vaccine based on a platform similar to the well-characterized Tag/Catcher-AP205 derived technology[23,24]. Accordingly, a split-protein Tag/Catcher system[25–27] is used to conjugate and display the RBD of the SARS-CoV-2 spike protein on the protein surface of preassembled bacteriophage AP205 capsid-like particles (CLPs). Importantly, the modular Tag/Catcher-AP205 CLP vaccine design makes it possible to replace the current vaccine antigen relatively quickly in the event that the SARS-CoV-2 virus should acquire mutations in the RBD domain reducing the efficacy of an existing vaccine. CLPs are supramolecular structures assembled from multiple copies of a single viral coat protein, thus resembling the structure of the virus from which they are derived[28]. Importantly, CLPs are considered safe, as they do not contain any viral material and thus cannot infect or replicate[29]. Their resemblance with native viruses make them highly immunogenic, with important immunogenic features like their size (enabling direct draining to the lymph nodes) and their repetitive surface epitope-display[30–33]. In fact, many preclinical studies have shown that high-density and unidirectional antigen-display on CLPs consistently increase the immunogenicity of the presented antigen, and promote strong and durable antigen-specific antibody responses[34,35]. Importantly, the immune activating properties of the repetitive CLP epitope-display appear to be universally recognized in all mammalian species, including humans[36,37]. Indeed, a strong proof-of-concept in humans has been established by the Human Papillomavirus (HPV) VLP vaccines (Cervarix®, Gardasil®, and Gardasil 9®), which appear to generate lifelong protective antibody responses after a single immunization[38–40]. Finally, the production of AP205 CLPs in *E. coli* is highly scalable and results in encapsulation of bacterial RNA, which act as a potent Th1-type adjuvant through engagement of toll-like receptor (TLR) 7/8[41].

Here, we describe the design, development, and immunogenicity in mice of two CLP-based SARS-CoV-2 RBD vaccines. Two RBD antigen designs were evaluated based on their stability and accessibility to the ACE2 receptor binding epitope, before and after coupling to CLPs. The immunogenicity of the vaccines were assessed in mice, and the neutralization capacity of vaccine-induced immunoglobulins were evaluated using two different clinical SARS-CoV-2 isolates. Together, these data establish a strong proof-of-concept for the CLP-RBD Covid-19 vaccine, which was highly immunogenic and elicited a strong viral neutralizing response. The potential ability of the CLP-platform to promote a strong and focused Th1-type antibody response targeting neutralizing epitopes on the RBD is promising, and supports the further clinical development of the RBD-CLP vaccine. We believe our RBD-CLP vaccine holds the potential to induce a protective immune response in humans, and thus, the lead RBD-CLP vaccine has been forwarded for GMP production and clinical development.

## Results

**Development and characterization of a CLP-based SARS-COV-2 vaccine**. The RBD (amino acids (aa) 319-591) of the SARS-CoV-2 spike protein (Sequence ID: QIA20044.1) was genetically fused at either the N- or C-terminus to the split-protein Catcher, used for conjugation to the CLP (Fig. 1a, c). The two RBD antigens (termed RBDn and RBDc, respectively) were expressed in Schneider-2 (ExpresS²) insect cells, yielding approximately 8 mg/L for transient cell line and 50 mg/L for stable cell line. RBDc appeared to be a high-quality monomeric protein (supplementary Fig. 1), and the same was true for RBDn. The split-protein peptide Tag was genetically fused to the coat protein of the AP205 and expressed in *E. coli* with yields in the gram per liter range. The recombinant Tag-AP205 protein spontaneously forms CLPs presenting the peptide Tag on its surface[23] (Fig. 1c). Mixing of Catcher-RBD and Tag-CLPs results in the formation of a covalent isopeptide bond between the Catcher and Tag[42–47]. Covalent coupling of the RBD antigens to the CLPs was confirmed by SDS-PAGE analysis, by the appearance of a protein band of 60 kDa, corresponding to the added size of the RBD antigen (43 kDa) and Tag-CLP subunit (16.5 kDa) (Fig. 1b, lane 2 and 5). The samples were subjected to a stability spin test (16000 g, 2 min), showing no loss of the coupling band (60 kDa), indicating that the vaccines are stable and not prone to precipitation or aggregation (Fig. 1b, lane 3 and 6). The coupling efficiency of the reactions were assessed by densitometry to be approximately 33% for the RBDc and 45% for the RBDn vaccine. For the RBDc-CLP and RBDn-CLP vaccines, this means that each CLP (build from 180 subunits) was decorated with ~60 RBDc and ~80 RBDn antigens, respectively. A higher coupling efficiency could not be obtained by increasing the molar excess of antigen, indicating steric hindrance on the CLP surface. The Tag/Catcher mediated conjugation results in unidirectional display of the RBD antigens, thus the positioning of the Catcher on the RBD could affect how the antigen is oriented on the CLP surface (Fig. 1c). However, structural modelling of the RBD-CLP vaccine suggests, that both the N- and C-terminus of the RBD antigen are in close proximity to the CLP surface (Fig. 1d), and that RBD has a similar orientation whether the catcher is attached N- or C terminally. In addition, the modelling suggests that the ACE2 binding epitope on RBD will be accessible for immune recognition on the CLPs (Fig. 1d). After removal of unbound RBD, the integrity and aggregation of the vaccines were analyzed by transmission electron microscopy (TEM) and dynamic light scattering (DLS). TEM analysis confirmed the presence of intact CLP-antigen complexes of the expected size for both vaccines (Fig. 2a–c). However, DLS analysis showed that the RBDc-CLP vaccine had propensity for aggregation, as indicated by a high polydispersity (Pd% ~30) and showed evidence of larger aggregates (Fig. 2b). In contrast, the RBDn-CLP vaccine showed little aggregation with a single peak of the expected size of monodisperse CLP antigen complexes (~50 nm) (Fig. 2d).

**Qualification of antigen structure and CLP-display**. The protein fold of the recombinant RBD antigens was validated by

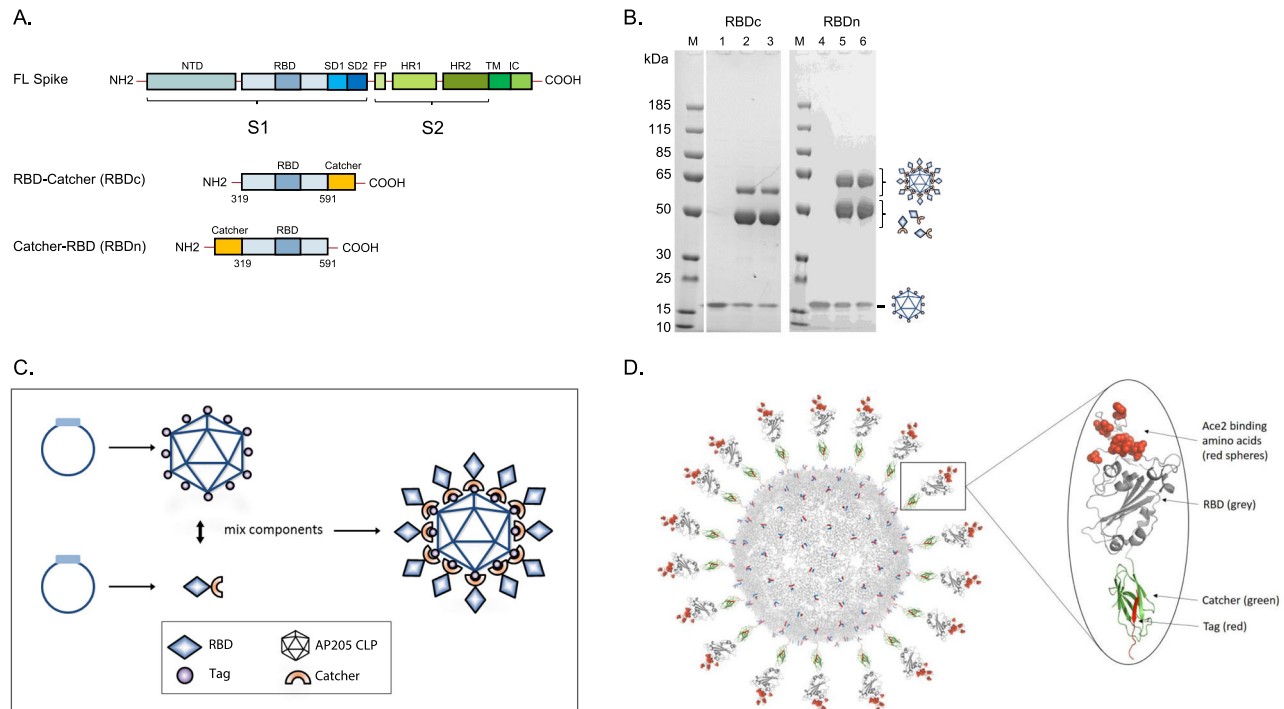

**Fig. 1 RBD-CLP vaccine design and characterization. a** Schematic representation of the complete SARS-CoV-2 spike protein including the two RBD-Catcher antigen designs. NTD = N-terminal domain, FL = full-length, RBD = receptor-binding domain, SD1 = subdomain 1, SD2 = subdomain 2, FP = fusion peptide, HR1 = heptad repeat 1, HR2 = heptad repeat 2, TM = transmembrane region, IC = intracellular domain. **b** Individual vaccine components on a reduced SDS-PAGE. M: marker, lane 1: unconjugated Tag-CLPs (16.5 kDa), lane 2: RBDc-CLP conjugation after overnight incubation at 4 °C (60 kDa), lane 3: RBDc-CLP conjugation after overnight incubation at 4 °C (60 kDa) + spin test, lane 4: unconjugated Tag-CLPs (16.5 kDa), lane 5: RBDn-CLP conjugation after overnight incubation at RT (60 kDa), lane 6: RBDn-CLP conjugation after overnight incubation at RT (60 kDa) + spin test. This test was repeated >5 times and showed consistent results. **c** Schematic representation of the Tag/Catcher-AP205 technology used to create the RBD-CLP vaccines. The genetically fused peptide Tag at the N-terminus of each AP205 capsid protein (total of 180 subunits per CLP) allows unidirectional and high-density coupling of the RBD antigens, via interaction with the N- or C-terminal Catcher (i.e. the corresponding binding partner). **d** Structural illustration of the RBD-CLP vaccines, based on the SARS-CoV-2 spike (Sequence ID: QIA20044.1), Tag/Catcher, and AP205 CLP (Sequence ID: NP_085472.1)[41] structures. The Tag is shown in red, Catcher in green, RBD in grey with the amino acids residues involved in ACE2 binding interface shown as red spheres.

measuring their affinity for binding to the human receptor, ACE2. Specifically, the binding affinity to ACE2 was measured for each antigen, before and after coupling to the CLP. Binding of RBDn was performed in a concentration titration series using an Attana Biosensor and showed high affinity binding to immobilized ACE2 with a $K_D$ of 19.4 nM (Fig. 3a). Similar binding kinetics were observed for both RBDc ($K_D = 34.6$ nM) and full-length SARS-CoV-2 spike ectodomain (Supplementary fig. 2a, b). This demonstrates that the native structure around the ACE2 binding epitope in the full-length spike protein is maintained when RBDn and RBDc are expressed as soluble proteins. Importantly, both RBD antigens bound effectively to the ACE2 receptor also when displayed on CLPs (Fig. 3b and Supplementary fig. 2C), thus confirming that the CLP display maintained exposure of the ACE2 binding epitope.

**Immunogenicity of the RBD-CLP vaccines**. The immunogenicity of the RBD-CLP vaccines (RBDn-CLP and RBDc-CLP) was assessed in BALB/c mice serum, obtained after prime and boost immunizations, and compared to the immunogenicity of vaccine formulations containing equimolar soluble antigen (RBDn and RBDc). All vaccines were formulated in Squalene-Water-Emulsion (Addavax™) adjuvant. Antigen-specific IgG titers were measured by ELISA using a recombinant full-length (aa35-1227) SARS-CoV-2 spike protein for capture. Both RBD-CLP vaccines led to seroconversion in all mice, and booster immunizations distinctly increased the antibody levels (Fig. 4a–d).

Furthermore, IgG levels were markedly higher in RBD-CLP vaccinated mice, compared to mice vaccinated with the soluble protein ($p = <0.05$, Fig. 4a–d). Antibody titers in sera from BALB/c mice immunized using a standard prime-boost regimen (5 µg of RBDn-CLP) were not significantly different from the titers shown in Fig. 4 (Supplementary fig. 3). Finally, the IgG subclass profile was assessed for the antibody responses induced by the soluble RBDn and RBDn-CLP vaccine. The soluble RBDn vaccine induces a Th2-biased immune response indicated by a predominant induction of IgG1 antibodies (Fig. 4f), while the corresponding RBDn-CLP vaccine induces a more balanced Th1/Th2-type response with significantly increased levels of IgG2a and IgG2b (Fig. 4e). This was further supported by FACS analysis showing statistically increased levels of RBD-specific CD4 + IFNg+ T cells in mice immunized with the RBDn-CLP vaccine (Supplementary fig. 4).

**Neutralization capacity of vaccine-induced anti-RBD antibodies**. The capacity of the vaccine-induced mouse antibodies to neutralize SARS-CoV-2 virus was measured in vitro by two different external laboratories, by testing the capacity of two different clinical SARS-CoV-2 isolates to infect humanized VeroE6 cells. Serum from mice immunized with RBDc-CLP showed significantly higher neutralization capacity than serum from mice immunized with soluble RBDc (Fig. 5a, supplementary fig. 5, performed by Aarhus University). Furthermore, after the first immunization with the RBDn-CLP vaccine, serum exhibited

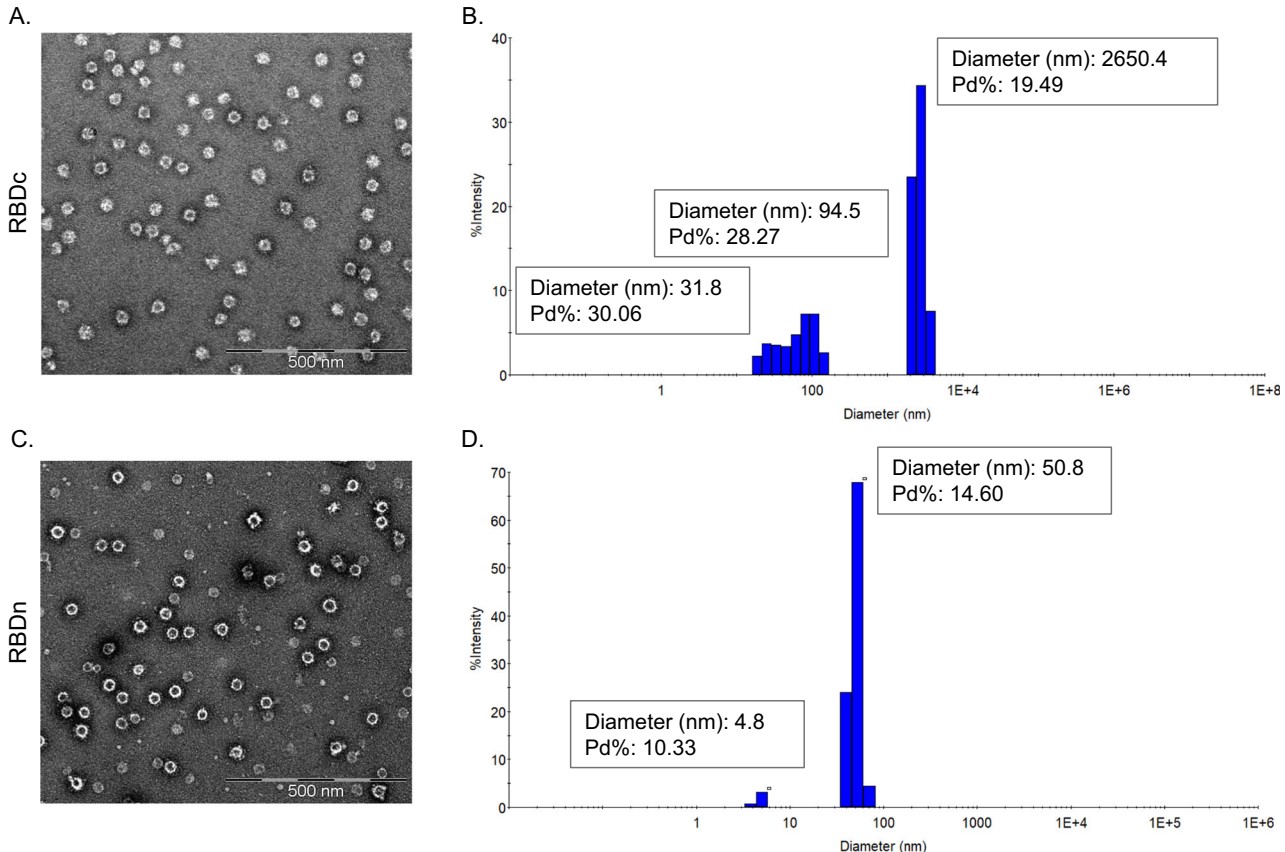

**Fig. 2 Vaccine quality assessment. a**, **c** Transmission electron microscope (TEM) images of the negatively stained purified RBDc-CLP or RBDn-CLP vaccine. Scale bar is 500 nm. **b**, **d** Histogram of the % intensity of the purified RBDc-CLP or RBDn-CLP particles from DLS analysis. Annotated are the average diameter and polydispersity (Pd%) for the particles.

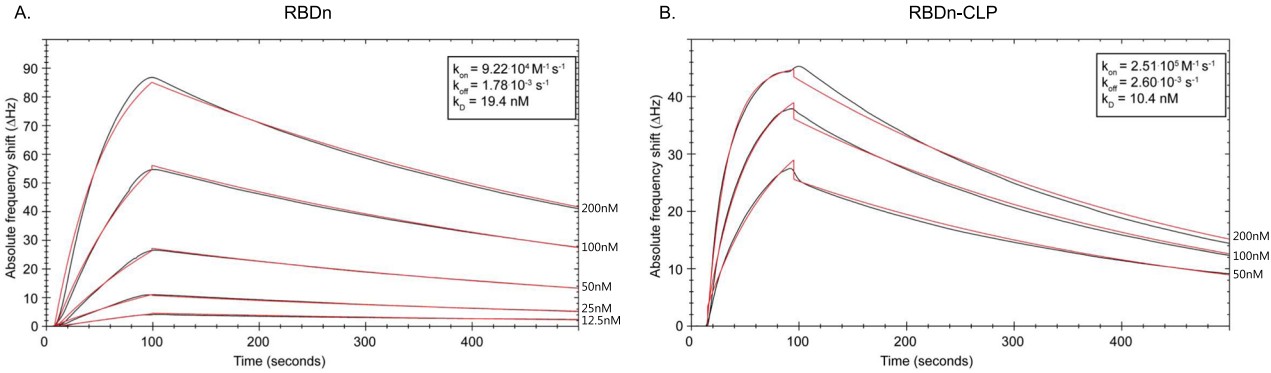

**Fig. 3 ACE2 binding kinetics for RBDn and RBDn-CLP. a** Binding of RBDn to immobilized hACE2. **b** Binding of ExpreS$^2$ produced ACE2 to immobilized RBDn-CLP. Real time binding (black curves) are fitted using a 1:1 simple binding model (red curve). Analyte concentrations are shown to the right and $k_{on}$, $k_{off}$, and $k_D$ are boxed.

above 50% neutralization titer at a serum dilution of 1:160 (Fig. 5b). Following booster immunizations, serum from these mice showed over 50% neutralization even at a dilution of 1:40960 (Fig. 5c, supplementary fig. 6). Similar results were obtained using a different clinical SARS-CoV-2 isolate, (Supplementary fig. 7, performed by Leiden University). A correlation analysis between the ELISA antibody titers and neutralization capacity, showed that there was a positive correlation between these measurements (Ks = 0.9583, $p$ = 0.0002) in mice immunized with the CLP vaccines, but not in the mice vaccinated with soluble RBD (Ks = 0.1991, $p$ = 0.6364) (Fig. 5d). The virus neutralization capacity was also evaluated for human serum from individuals having recovered from COVID-19 (Fig. 6a). Prior to this analysis, serum samples were grouped based on having either high or low SARS-CoV-2 binding antibody titers in ELISA capacity (i.e. >400 or ≤400 end-point titer, respectively) (supplementary fig. 8). Serum from mice receiving multiple immunizations with the RBDn-CLP vaccine showed markedly higher virus neutralization activity compared to the serum from any of the sera from patients recovered from COVID-19. However, serum from mice immunized once with RBDn-CLP showed similar neutralizing activity than the high patient sera (Fig. 6b). Samples from patients with high ELISA titers exhibited higher virus neutralization activity than samples from patients with low

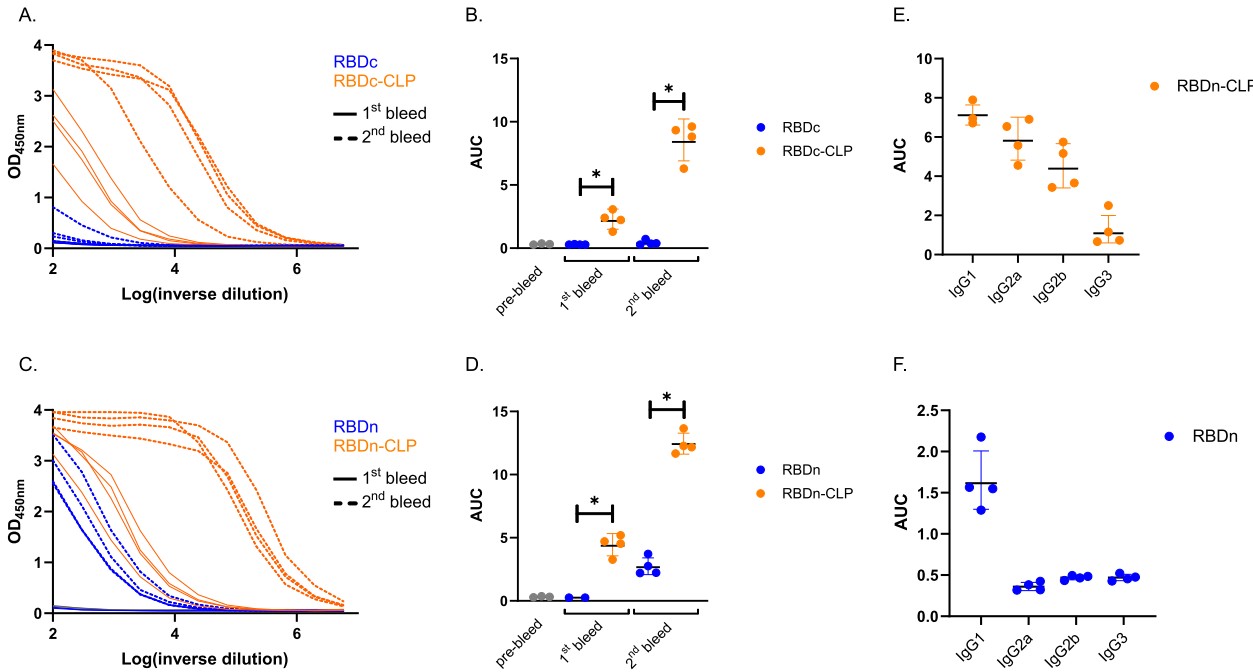

**Fig. 4 RBD-CLP vaccines induce high antigen-specific antibody titers in mice.** Serum samples were obtained before vaccination (pre-bleed) and two weeks after primary (1st bleed) and boost (2nd bleed) vaccination, respectively. ELISA results are depicted both as raw serum dilution curves (**a, c**) as well as the area under curve (AUC) and the geometric mean/SD (**b, d**). **a** Serum dilution curves and **b** geometric mean titer/SD (RBDc 1st bleed: GMT = 0.289; RBDc 2nd bleed: GMT = 0.4369; RBDc-CLP 1st bleed: GMT = 2.252; RBDc-CLP 2nd bleed: GMT = 8.515) of total anti-SARS-CoV-2 spike (aa35-1227) IgG antibodies detected in sera from BALB/c mice immunized intramuscularly with soluble RBDc (prime 2 μg/boost 2 μg) (n = 4) or CLP-displayed RBDc (RBDc-CLP) (prime 1 μg/boost 1 μg) (n = 4). **c** Serum dilution curves and **d** geometric mean titer/SD (RBDn 1st bleed: GMT = 0.2578; RBDn 2nd bleed: GMT = 2.727; RBDn-CLP 1st bleed: GMT = 4.427; RBDn-CLP 2nd bleed: GMT = 12.44) of total anti-SARS-CoV-2 spike (aa35-1227) IgG antibodies detected in sera from Balb/c mice immunized intramuscularly with soluble RBDn (prime 5 μg/boost 5 μg) (n = 4) or CLP-displayed RBDn (RBDn-CLP) (prime 6.5 μg/boost <0.1 μg/boost 6.5 μg) (n = 4). **e** IgG subclass profile in sera from mice (n = 4) immunized (prime-boost) with the RBDn-CLP vaccine. IgG1 GMT = 7.108; IgG2a GMT = 5.814; IgG2b GMT = 4.390; IgG3 GMT = 1.088. **f** IgG subclass profile in sera from mice (n = 4) immunized (prime-boost) with the soluble RBDn vaccine. IgG1 GMT = 1.615; IgG2a GMT = 0.3586; IgG2b GMT = 0.4696; IgG3 GMT = 0.4690.

ELISA titers ($p = 0.0025$) (Fig. 6b). Together these data establish a strong proof-of-concept for the capacity of the RBDn-CLP vaccine to elicit a strong antibody response targeting neutralizing epitopes in the RBD of the SARS-CoV-2 spike protein.

## Discussion

In less than 1 year, more than 35 million confirmed cases of SARS-CoV-2 infection, and more than 1 million COVID-19 related deaths have been reported[48]. Thus, development of an effective vaccine is of high priority worldwide. The ideal SARS-CoV-2 vaccine should be safe, and capable of activating a long-term protective immune response. High immunogenicity is pivotal for vaccine efficacy and represents a fundamental challenge for vaccine development[49]. In the context of COVID-19, the elderly carry an increased risk of serious illness[50], but it is also well known that this group generally responds less effectively to vaccination[51,52]. In addition, the balance between immunogenicity and safety vary among different vaccine platforms, and concerns have been raised that some SARS-CoV-2 vaccines can potentially cause enhanced disease. This risk is believed to be higher for vaccines which fail to induce a sufficiently strong virus neutralizing antibody responses[53]. Although it is still unclear whether natural infection with SARS-CoV-2 can induce long-term protective immunity, natural infection with members of the coronavirus family causing common cold, provide only short-term protection[54–56]. Accordingly, COVID-19 vaccines may need to induce a stronger and more durable effective immune response than natural infection, in order to provide long term protection.

Our strategy for developing a CLP-based COVID-19 vaccine displaying the SARS-CoV-2 spike RBD holds several potential advantages. Firstly, other CLP-based vaccines have shown to be safe and highly immunogenic in humans. In fact, the marketed Human Papillomavirus (HPV) vaccines, based on HPV L1 VLP, induce potent and durable antibody responses otherwise only seen after vaccination with live-attenuated viral vaccines[38–40]. With regard to safety, several experts have stated that SARS-CoV-2 vaccines should preferentially induce a high level of neutralizing antibodies, while avoiding activation of Th2 T cells, to reduce the risk of eosinophil-associated immunopathology following infection after SARS-CoV-2 vaccination[53,57]. To this end, it seems ideal that production of AP205 CLPs in *E. coli* results in encapsulation of bacterial host cell RNA, promoting Th1 type responses by activation of TLR7/8[41]. Additionally, a recent review[49], comparing different SARS-CoV-2 vaccine candidates, suggests that recombinant proteins and nanoparticles are the preferred option for obtaining high safety, high immunogenicity and hold potential for raising neutralizing antibody titers. Thus, the strategy of targeting only the RBD of the SARS-CoV-2 spike protein, along with the unique ability of the Tag/Catcher-AP205 platform to present the RBD in a high-density and unidirectional manner, may not only ensure high immunogenicity, but may also enable induction of responses with a high proportion of neutralizing compared to binding antibodies[7–12,58]. In fact, the unidirectional antigen display enabled by the Tag/Catcher-AP205 platform has previously been exploited to selectively favor induction of antibodies targeting desired epitopes[59]. It is thus encouraging that both our

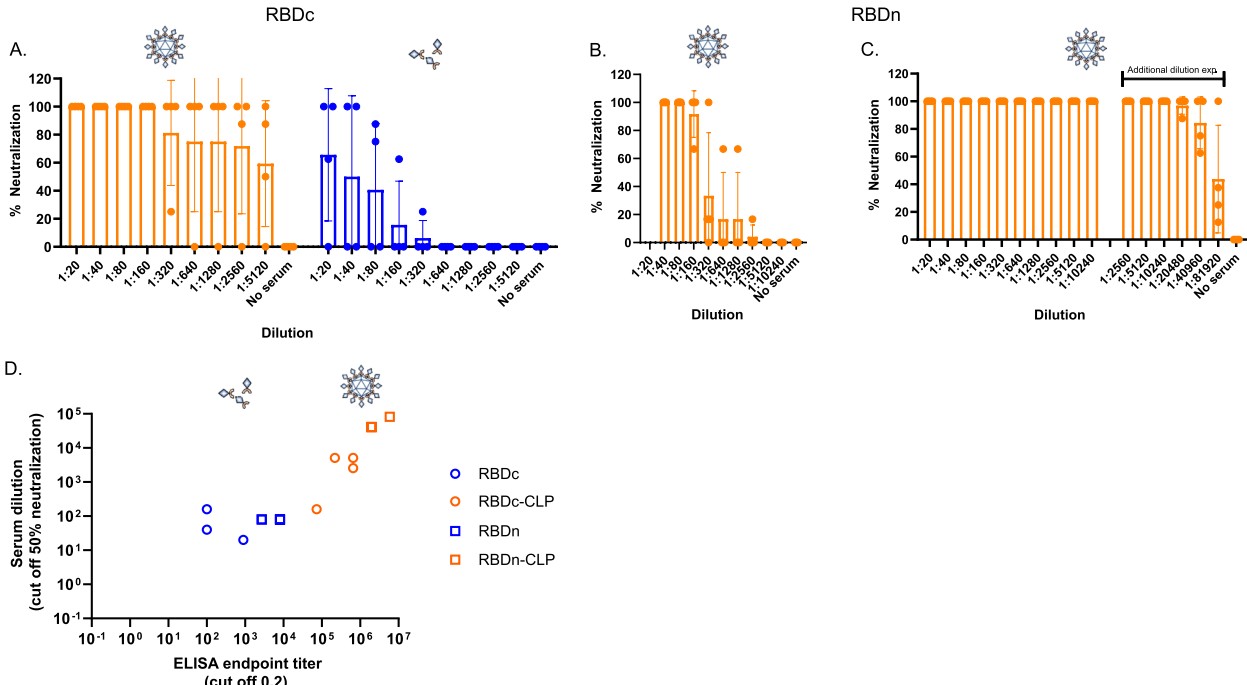

**Fig. 5 Serum from mice immunized with RBD-CLP vaccines neutralizes SARS-CoV-2 in vitro. a** Serum from groups of mice ($n = 4$) immunized (prime 1 ug/boost 1 ug) with RBDc-CLP (orange) ($n = 4$) or soluble RBDc (prime 2 ug/boost 2 ug) (blue) was mixed with a SARS-CoV-2 virus and tested for cell entry. Each dot represents the percentage neutralization per mouse per dilution. Bars represent the mean and standard deviation. **b, c** Serum from groups of mice ($n = 4$) immunized (prime 6.5 μg/boost <0.1 μg/boost 6.5 μg) with RBDn-CLP obtained after the first (**b**) or last (**c**) immunization was mixed with a SARS-CoV-2 virus and tested for cell entry. Each dot represents the percentage neutralization per mouse per dilution. Bars represent the mean and SD. **d** Correlation between ELISA IgG endpoint titer (2nd bleed, cutoff 0.2) and the serum dilution required for 50% virus neutralization. The ELISA endpoint titers and the 50% neutralization titers were measured on the same set of serum samples obtained from groups of mice ($n = 4$) immunized with RBDc ($n = 4$), RBDc-CLP ($n = 4$), RBDn ($n = 4$) or RBDn-CLP ($n = 4$) (Fig. 4a–c). Each dot represents one mouse. A Pearson r correlation was used to assess the relationship between variables.

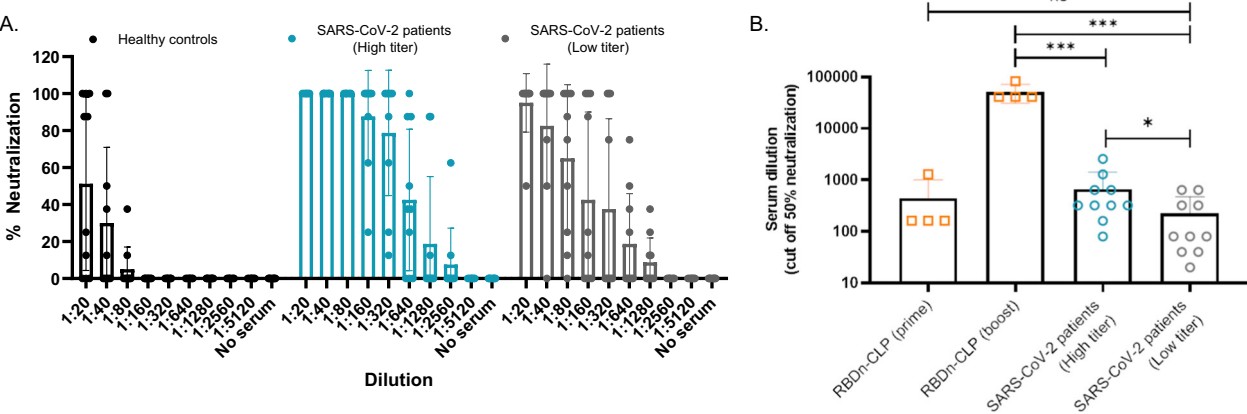

**Fig. 6 Neutralization capacity of serum from convalescent SARS-CoV-2 patients, performed at Aarhus University. a** A dilution series of individual human plasma samples from SARS-CoV-2 patients (with either high ($n = 5$) or low ($n = 5$) ELISA binding titers against recombinant SARS-CoV-2 spike protein) or healthy controls ($n = 5$) were mixed with a clinical SARS-CoV-2 isolate and tested for cell entry. Each dot represents the percentage neutralization per sample, per dilution. Bars represent the mean of the group with a standard deviation. **b** Endpoint serum dilution required for 50% virus neutralization. Each dot represents the serum dilution needed for 50% virus neutralization according to the dilution titration of the sera in the neutralization assay (Fig. 6a and Fig. 5b, c). Bars represent the mean of the group with a standard deviation. One sided non-parametric Mann-Whitney test was used for statistical comparison. Statistically significant differences are marked by asterisk: ns=non-significant, *: $p < 0.05$**: $p \leq 0.005$, ***: $p \leq 0.001$. Specifically, *** (RBDn-CLP(boost) vs SARS-Cov-2 patients (high titer)) $p = 0.001$; *** (RBDn-CLP(boost) vs SARS-Cov-2 patients (low titer)) $p = 0.001$; * (SARS-Cov-2 patients (high titer) vs SARS-Cov-2 patients (low titer)) $p = 0.0286$.

RBD-CLP vaccine candidates appear to expose the ACE2 binding epitope, as evidenced by the strong binding of RBD-CLP complexes to ACE2. Our data, comparing the immunogenicity of soluble versus CLP-displayed RBD antigens in mice, show a remarkable effect of the CLP display (approx. 3- and 4-fold difference for the RBDc and RBDn vaccine for the 2nd bleed, respectively). Indeed, the observed low intrinsic immunogenicity of the soluble RBD antigen, even in the presence of Addavax[TM] adjuvant, emphasizes the need of an effective vaccine delivery platform, and raises concern whether vaccines based on soluble recombinant proteins will be sufficiently immunogenic in humans. Additionally, our data show that delivery of the RBD antigen by the Tag/Catcher-AP205 platform promotes induction of IgG2a and IgG2b subclasses (i.e. characteristic of a Th1 response), which is believed to reduce the potential risk of vaccine-related enhancement of disease[60]. Further analysis of the neutralizing capacity of vaccine-induced mouse antibodies shows that RBD-CLP vaccines also elicite antibody responses with significantly higher neutralization capacity. This result may not only be due to increased immunogenicity of the CLP-displayed RBD antigen, but could also reflect a higher proportion of neutralizing antibodies in the total pool of vaccine-induced antibodies. Indeed, a strong positive correlation is observed between vaccine-induced antibody titers and virus neutralization activity among the RBD-CLP immunized mice. A similar correlation is not seen for the soluble RBD vaccines. As suggested in the literature, it is expected that a correlation between binding antibodies and neutralizing antibodies indicates a reduced risk of enhanced disease, as it was seen with SARS-CoV-2 patients[57]. Serum samples from convalescent patients show similar neutralization titers as those measured for mouse sera obtained after a prime immunization of RBDn-CLP. A recent review, compiling all the latest data on SARS-CoV-2 vaccine development, suggests that a > 50% neutralizing titers at an endpoint titer dilution of 100-500 would be needed to confer protection[49]. In relation to this, our RBDn-CLP vaccine induces over 50% neutralization at a dilution of 40960 serum dilution, suggesting that it could have the potential to trigger a robust immune response in humans. To this date, many studies have shown that both genetic and protein-based vaccines need to be supported by a stronger vaccine platform or adjuvant to enable sufficiently potent immune responses[61,62]. Indeed, when looking at emerging data on SARS-CoV-2 vaccine development, it appears that the vaccines that are fast to produce (i.e., genetic and viral vectors) might not be able to elicit antibody titers sufficient to confer long-lived protection[49]. Additionally, vaccines have many times failed due to low immunogenicity when testing in human clinical trials, despite having produced encouraging results in preclinical models[63,64]. Thus, in the case of SARS-CoV-2, it seems that recombinant proteins or killed/attenuated virus vaccines would most likely be the ones enabling responses strong enough for long-term protection[49]. However, killed or live-attenuated viruses have potential safety concerns. Thus, we propose that the Tag/Catcher-AP205 system is a good platform for delivery of the RBD antigen, to enable induction of a strong, long-lasting and highly neutralizing antibody response, while avoiding high safety risks. Specifically, the intrinsic CLP properties provide the balance between high immunogenicity and safety, which is of main importance for a vaccine supposed to protect globally, including the at risk populations. Additionally, the high-density coupling of the RBD antigen to the surface of AP205 CLP is expected to limit anti-CLP immunity from potentially posing a negative effect on the induction of antigen-specific antibody responses upon repeated booster vaccinations[65,66]. Based on these results, the RBDn-CLP vaccine has been selected as our lead candidate, due to its high stability and low aggregation compared to RBDc-CLP, as well as its high immunogenicity and neutralizing capacities, in mice. Thus, this vaccine has been transferred to GMP, with a planned phase 1 clinical testing (funded by H2020).

## Methods

**Design, expression, and purification of recombinant proteins**. RBD antigens were designed with boundaries aa319-591 of the SARS-CoV-2 sequence (Sequence ID: QIA20044.1). The RBD antigens were genetically fused with the split-protein Catcher at the N-terminus or the C-terminus (referred to as RBDn and RBDc, respectively). Both antigen constructs had an N-terminally BiP secretion signal and a C-terminal C-tag (N-RBD-EPEA-C) used for purification. A GSGS linker was inserted between the RBD and the Catcher. The final gene sequences were codon optimized for expression in *Drosophila melanogaster* and were synthesized by Geneart©. The ExpreS2 platform was used to produce all proteins by transient transfection. Briefly, Schneider-2 (ExpreS2) cells were transiently transfected using transfection reagent (ExpreS2 Insect TRx5, ExpreS2ion Biotechnologies) according to manufacturer's protocol. Cells were grown at 25 °C in shake flasks for 3 days before harvest of the supernatant containing the secreted protein of interest. Cells and debris were pelleted by centrifugation (5000 rpm for 10 min at 4 °C) in a Beckman Avanti JXN-26 centrifuge equipped with a JLA 8.1000 swing-out rotor. The supernatant was decanted and passed through a 0.22 μm vacuum filter (PES) before further processing. The supernatant was passed over a Centramate tangential flow filtration (TFF) membrane (0.1m², 10 kDa MWCO, PALL) mounted in a SIUS-LS filter holder atop a SIUS-LS filter plate insert (Repligen/TangenX). The retentate was concentrated ten-fold by recirculation through a concentration vessel of 1 litre volume without stirring. Buffer exchange was performed by diafiltration until achieving a turn-over-volume of 10.

The crude protein was loaded onto a Capture Select C tag resin (Thermo Fisher) affinity column and washed with capture buffer (25 mM Tris-HCl, 100 mM NaCl, pH7.5). The captured protein was step-eluted in 25 mM Tris-HCl (pH7.5) containing increasing concentrations of MgCl₂ (0.25 M, 0.5 M, 1 M and 2 M). Data were collected on Unicorn software (Citivalifesciences, Marlborough, USA, version 5.11) and fractions containing the protein of interest were pooled and concentrated (Amicon 15 ml, 10 kDa or 30 kDa MWCO). Concentrated protein was loaded onto a preparative Superdex-200pg 26/600 (Cytiva) SEC column equilibrated in 1x PBS (Gibco) and eluted in the same buffer. Fractions containing the monomeric RBD protein were pooled and concentrated as above. The ACE2 protein (aa1–615) and the spike protein (aa.35-1227)-Ctag (ΔTM-ΔFurin-CoV-PP-Ctag)) were N-terminally tagged with a BiP secretion signal and a C-terminal Twin-Strep-tag (Iba, GmbH) affinity-tag. The crude protein was loaded onto a StreptactinXT (IBA) affinity column. Proteins were eluted using capture buffer (100 mM Tris-HCl, 150 mM NaCl, 1 mM EDTA pH 8.0) supplemented with 50 mM D-Biotin (BXT buffer, Iba GmbH)

**Design, expression, and purification of Tag-CLP**. The proprietary peptide-binding Tag and a linker (GSGTAGGGSGS) was added to the N-terminus of the *Acinetobacter phage* AP205 coat protein (Gene ID: 956335). The gene sequence was inserted into the pET28a(+) vector (Novagen) using NcoI (New England Biolabs) and NotI (New England Biolabs) restriction sites. The Tag-CLP was expressed in BL21 (DE3) competent *E. coli* cells (New England Biolabas) according to manufacturer's protocols, and purified as described below for the CLP vaccines.

**Formulation and purification of the RBD-CLP vaccines**. The Tag-CLP and the RBDc antigen were mixed in a 1:2 molar ratio in 100 mM Bis-Tris, 250 mM NaCl (pH 6.5) buffer overnight at 4 °C. Tag-CLP and RBDn antigen were mixed in a 1:1 molar ratio in 1xPBS, 5% glycerol and incubated overnight at room temperature. Different working buffers for RBDn and RBDc vaccines were selected according to a buffer screen to ensure vaccine stability. A subsequent buffer screen showed that the RBDn-CLP was stabilized by the addition of different sugars (sucrose, xylitol and trehalose). Accordingly, PBS buffer, pH 7.4, supplemented by 400 mM xylitol was chosen for quality assessment of the RBDn vaccine. The mixture of RBD and CLP was subjected to a spin test to assess stability. Specifically, a fraction of the sample was spun at 16000 g for 2 min, and equal amounts of pre- and post-spin samples were subsequently loaded on a reduced SDS-PAGE to assess potential loss in the post-spin sample due to precipitation of aggregated RBD-CLP complexes. The RBD-Catcher coupling efficiency was calculated as percentage conjugation (i.e., number of bound antigens divided by the total available binding sites (=180) per CLP) by densitometric analysis of on the SDS-PAGE gel, using ImagequantTL, as previously described[67]. In parallel, RBDc-CLP was purified by density gradient ultracentrifugation by adding the RBDc-CLP onto an Optiprep™ step gradient (23, 29 and 35%) (Sigma-Aldrich) followed by centrifugation for 3.30 h at 47800 rpm. The conjugated RBDn-CLP was purified by dialysis (cutoff 1000 kDa) in a 1xPBS with 5% (v/v) glycerol for immunization studies or 400 mM xylitol for quality assessment.

**Quality assessment of the RBD-CLP vaccines**. Purified RBD-CLP were both quality checked by negative stain Transmission electron microscopy (TEM) (detailed description 10.1038/s41598-019-41522-5) as well as by Dynamic Light Scattering (DLS) analysis (DynaPro Nanostar, Wyatt technology). For DLS analysis, the RBD-CLP sample was first spun at 21,000 g for 2.5 min and then loaded into a disposable cuvette. The sample was then run with 20 acquisitions of 7 sec each. The estimated diameter of the RBD-CLP particle population and the percent polydispersity (%Pd) was calculated by Wyatt DYNAMICS software (v7.10.0.21, US).

**ACE2 binding kinetics by Attana© Biosensor**. Kinetic interaction experiments of RBD antigens and CLP-RBD binding to hACE2 were performed using a biosensor QCM Attana A200 instrument (Attana AB) and data were collected on Attache Office 2.1. hACE2 (50 μg/ml) or VLP-RBDn (50 μg/ml) were immobilized on a LNB carboxyl chip by amine coupling using EDC and S-NHS chemistry following manufacturer's instructions. A non-coated LNB chip was used as reference. Two-fold dilution series of RBDc (200nM-6.25 nM) and RBDn (200nM-12.5 nM) were prepared in 1xPBS pH 7.4. ExpreS$^2$ produced hACE2 (200nM-50nM) was prepared in 1xPBS + 400 mM xylitol pH7.4 running buffer. All sensorgrams were recorded at 25 _μl/min at 22 °C using an 84 s association and 3000 s dissociation time to allow complete baseline recovery. The absolute change in frequency (ΔHz) during association and dissociation were analyzed using Attester Evaluation software (Attana AB). Injection of running buffer (background binding) was subtracted for each sensorgram prior to fitting $k_{on}$ and $k_{off}$. The kinetic parameters were calculated using a 1:1 binding model using TraceDrawer software (Ridgeview Instruments AB).

**ACE2 binding to RBD-CLP by ELISA**. RBDc-CLP binding to ACE2 was performed using an enzyme-linked immune-sorbent assay (ELISA). 96-well plates (Nunc MaxiSorp) were coated overnight at 4 °C with 0.05 μg/well recombinant ACE2 produced in ExpreS$^2$ cells. Plates were blocked for 1 h at room temperature (RT) using 0.5% skimmed milk in PBS. 2.5 ug purified RBDc-CLP was added per well, or CLP alone and RBD alone as controls and incubated for 1 h at RT. Plates were washed three times in PBS between each step. Mouse monoclonal antibody (produced in-house), detecting AP205 was diluted 1:10,000 in blocking buffer, followed by incubation for 1 h at RT. Horseradish peroxidase (HRP) conjugated goat anti-mouse IgG (Life technologies, A16072) was diluted 1:1000 in blocking buffer followed by 1 h incubation at RT. Plates were developed with TMB X-tra substrate (Kem-En-Tec, 4800 A) and absorbance was measured at 450 nM. Data were collected on a BioSan HiPo MPP-96 microplate readerand analyzed using GraphPad Prism (San Diego, USA, version 8.4.3).

**Mouse immunization studies**. Experiments were authorized by the Danish National Animal Experiments Inspectorate (Dyreforsøgstilsynet, license no. 2018-15-0201-01541) and performed according to national guidelines. Mice were kept in rooms at a temperature of 22 ºC (±2 ºC), with a humidity of 55% (±10%), air in the room was changed 8–10 times/hour, according to Danish animal experiments regulations (bekendtgørelse n12 from 07.01.2016). 12–14 weeks old female BALB/c mice (Janvier, Denmark) were immunized intramuscularly, in the thigh, with either 2 μg free RBDc antigen (1x PBS, pH7.4) (n = 4) or 1 μg CLP-displayed RBDc (PBS with Optiprep™) (n = 4), using a two-week interval prime-boost regimen. For the RBDn study, mice were immunized with a dose of 5 μg free RBDn antigen (1x PBS, pH7.4) or 6.5 μg CLP-displayed RBDn (1xPBS, pH7.4, 5% glycerol) (n = 4) and boosted 2 weeks later with 5 μg free RBDn antigen (1x PBS, pH7.4) or 0.1 μg CLP-displayed RBDn (1xPBS, pH7.4, 5% glycerol) (n = 4). Considering the low dose used for the RBDn-CLP boost, it was decided to give them an extra boost a week later (3 weeks post prime) with 6.5 μg CLP-displayed RBDn (1xPBS, pH7.4, 5% glycerol) (n = 4). For both studies, the concentration of the antigen displayed on the CLP was calculated by densitometric measurement (ImageQuant TL), using a protein concentration ladder as a reference. All vaccines were formulated using Addavax™ (Invivogen). Blood samples were collected prior to the first immunization (pre-bleed) as well as two weeks after each immunization. Serum was isolated by spinning twice the blood samples down for 8 min at 800 g, 8 ºC.

**Analysis of vaccine-induced antibody responses**. Antigen-specific total IgG titers were measured by ELISA. 96-well plates (Nunc MaxiSorp) were coated overnight at 4 °C with 0.1 μg/well recombinant ExpreS$^2$ produced SARS-CoV-2 Spike (35-1227) protein in PBS. Plates were blocked for 1 h, RT using 0.5% skimmed milk in PBS. Mouse serum was diluted 1:100 in blocking buffer, and added to the plate in a 3-fold dilution, followed by incubation for 1 h at RT. Plates were washed three times in PBS in between steps. In order to measure total serum IgG, Horseradish peroxidase (HRP) conjugated goat anti-mouse IgG (Life technologies, A16072) was diluted 1:1000 in blocking buffer followed by 1 h incubation at RT. To measure IgG subclass, HRP goat anti-mouse IgG1 (Invitrogen, A10551), IgG2a (Invitrogen, M32207), IgG2b (Invitrogen, M32407) and IgG3 (thermofisher, M32707) were diluted 1:1000 in blocking buffer and incubated for 1 h at RT. Plates were developed with TMB X-tra substrate (Kem-En-Tec, 4800 A) and absorbance was measured at 450 nM. Data were collected on a BioSan HiPo MPP-96 microplate reader and analyzed using GraphPad Prism (San Diego, USA, version 8.4.3).

**Human serum collection and screen**. Study of samples from individuals recovered from Covid-19 infection for validation of serological SARS-CoV-2 assays was approved by the Regional Scientific Committee for the Capital Region of Denmark (H-20028627). Blood donors were asked for consent to use archive samples for use in the validation of new methods and assay investigations as quality control projects. Samples from SARS-CoV-2 convalescent individuals were obtained from a variety of convalescent patients in the Capital Region of Denmark with a confirmed SARS-CoV-2 NAAT result: The NAAT results were identified in the Danish Microbiology Database (MiBa) from February 2020 to

April 2020. Samples from 150 individuals bled on May 3rd were included in a national validation study of SARS-CoV-2 antibody immunoassays, of these, 20 samples were randomly selected. Antigen-specific total IgG titers were measured by ELISA. 96-well plates (Nunc MaxiSorp) were coated overnight at 4 °C with 0.1 μg/well recombinant ExpreS$^2$ produced SARS-CoV-2 Spike (35-1227) protein in PBS. Plates were blocked for 1 h, RT using 0.5% skimmed milk in TSM buffer (150 mM NaCl, 2 mM $CaCl_2$, 2 mM $MgCl_2$). Serum was diluted 1:50 in blocking buffer, and added to the plate in a 2-fold dilution, followed by incubation for 1 h, RT. Plates were washed three times in PBS in between steps. In order to measure total serum IgG, anti-human IgG-HRP (Dako, P0214) was diluted 1:4000 in blocking buffer followed by 1 h incubation at RT. Plates were developed with TMB plus 2 substrate (Kem-En-Tec, 4395 A) and absorbance was measured at 450 nM. Data were collected on a BioSan HiPo MPP-96 microplate reader and analyzed using GraphPad Prism (San Diego, USA, version 8.4.3). Serum were consequently put in 2 groups on behalf of high and low positive ELISA signals (i.e., >400 or ≤400 end-point titer, respectively).

**Virus Neutralization assay (University of Aarhus, Denmark)**. SARS-CoV2, Freiburg isolate, FR-4286 (kindly provided by Professor Georg Kochs, University of Freiburg) was propagated in VeroE6 expressing human TMPRSS2 (VeroE6-hTMPRSS2) (kindly provided by Professor Stefan Pöhlmann, University of Göttingen)[68] with a multiplicity of infection (MOI) of 0.05. Supernatant containing new virus progeny was harvested 72 h post infection, and concentrated on 100 kDa Amicon ultrafiltration columns (Merck) by centrifugation for 30 min at 4000 g. Virus titer was determined by TCID$_{50}$ assay and calculated by Reed-Muench method[69]. Sera from immunized mice or human serum/plasma (kindly provided by Herlev Hospital and Rigshospitalet, Denmark) were heat-inactivated (30 min, 56 °C), and prepared in a 2-fold serial dilution in DMEM (Gibco) + 2% FCS (Sigma-Aldrich) + 1% Pen/Strep (Gibco) + L-Glutamine (Sigma-Aldrich). Sera were mixed with SARS-CoV-2 at a final titer of 100 TCID$_{50}$/well, and incubated at 4˚C overnight. A no serum and a no virus (uninfected) control samples were included. The following day virus:serum mixtures were added to 2 × 104 Vero E6 TMPRSS2 cells seeded in flat-bottom 96-well plates, and incubated for 72 h in a humidified $CO_2$ incubator at 37 ˚C, 5% $CO_2$, before fixing with 5% formalin (Sigma-Aldrich) and staining with crystal violet solution (Sigma-Aldrich). The plates were read using a light microscope (Leica DMi1) with camera (Leica MC170 HD) at 4x magnification, and cytopathic effect (CPE) scored.

**Virus Neutralization assay (University of Leiden, Netherlands)**. SARS-CoV-2 (Leiden-001 isolate, unpublished) was propagated and titrated in Vero E6 cells [CRL-1580, American Type Culture Collection (ATCC)] using the tissue culture infective dose 50 (TCID$_{50}$) endpoint dilution method and the TCID$_{50}$ was calculated by the Spearman-Kärber algorithm[70]. Neutralization assays against live SARS-CoV-2 were performed using the virus micro-neutralization assay. Briefly, Vero-E6 cells were seeded at 10000cells/well in 96-well tissue culture plates 1 day prior to infection. Serum samples were heat-inactivated at 56 °C for 30 min and prepared in a 2-fold serial dilution (1:10-1280) in 60 μL EMEM (Lonza) supplemented with 1% pen/strep (Sigma-Aldrich, P4458), 2mM L-glutamine (PAA) and 2% FCS (Bodinco BV). Diluted sera were mixed with equal volumes of $_{120}$ TCID$_{50/60 μL}$ SARS-CoV-2 and incubated for 1 h at 37 °C. The virus:serum mixtures were then added onto Vero-E6 cell monolayers and incubated at 37 °C in a humidified atmosphere with 5% $CO_2$. Cells either unexposed to the virus or mixed with $_{120}$ TCID$_{50/60 μL}$ SARS-CoV-2 were used as negative (uninfected) and positive (infected) controls, respectively. 3 days post-infection, cells were fixed and inactivated with 40 μL 37% formaldehyde/PBS solution/well overnight at 4 °C. Cells were then stained with crystal violet solution 50 μL/well, incubated for 10 min and rinsed with water. Dried plates were evaluated for viral cytopathic effect and the serum neutralization titers were determined as the reciprocal value of the highest dilution resulting in complete inhibition of virus-induced cytopathogenic effect. For the purpose of graphical representation, samples with undetectable antibody titers were assigned values two-fold lower than the lowest detectable titer (titer 10), which corresponds to the nearest dilution that could not be measured (titer 5). A SARS-CoV-2 back-titration was also included with each assay run to confirm that the dose of the used inoculum was within the acceptable range of 30 to 300 TCID$_{50}$.

**Analysis of T cell responses after vaccination**. In order to measure specific T cell responses, BALB/C mice (n = 4) were immunized intramuscularly in a prime-boost-regime with 5 ug RBDn-CLP. 2 weeks post boost, spleens were harvested, and lymphocytes were incubated with a pool of peptides at a concentration of 1 uM, in presence of monensine (4 uM) at 37 °C, 5% $CO_2$ for 5 h. The peptide pool includes 16mer peptides with 10 amino acids overlap covering positions 343 to 436 of the SARS-CoV2 Spike protein. After incubation, cells were washed and stained for surface markers (CD4-PE-Cy7 and CD44-FITC) at a dilution of 1:100. Cells were then washed, fixed using paraformaldehyde and permeabilized using Saponin for intracellular staining (IFN-γ-APC). Finally, cells were washed and data was collected using a Fortessa 3-laser instrument (BD Biosciences) and DIVA software (BD FACSDIVA software v8.0.1). Data were analyzed using FlowJo software (v10.6.1, Tree Star, Ashland, OR).

**Reporting summary**. Further information on research design is available in the Nature Research Reporting Summary linked to this article.

## Data availability
The data that support the findings of this study are available from Bavarian Nordic but restrictions apply to the availability of these data, which were used under license for the current study, and so are not publicly available. Source data are provided with this paper and are available from the authors upon reasonable request and with permission of Bavarian Nordic. Accession codes are the following, SARS-CoV-2 spike protein (Sequence ID: QIA20044.1), *Acinetobacter phage* AP205 coat protein (Gene ID: 956335).

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

## Acknowledgements

The authors would like to express their deep gratitude to Nahla Chehabi, Andreas Frederiksen, Benjamin Jacobsen, Elham Marjan Mohammad Alijazaeri, Ana Maria Guzunov, Tenna Gribfeldt Jensen, and Ditte Rahbæk Boilesen for their excellent technical assistance. Furthermore, we would like to thank Blanca Lopez Mendez at the Biophysics Facility—Protein Structure and Function Program from Center for Protein Research, Copenhagen, for her assistance with the DLS measurement and analysis, as well as the Core Facility for Integrated Microscopy, Faculty of Health and Medical Sciences, University of Copenhagen for their excellent facilities and support in acquiring TEM images. The authors would like to thank the flow cytometry and single cell core facility at Copenhagen University for their support and assistance. We would like to acknowledge the IT, Substrate, Logistics, and Security Departments at the Faculty of Health and Medical Sciences, University of Copenhagen for ensuring that this important work in the laboratory could continue during times of total lockdown. Novo Nordisk and Merck Millipore are thanked for their continuous and direct support of process development. Attana A/S, BioradChemometec, Eppendorf, Hamilton, Thermo Fisher scientific, and Wyatt are thanked for their swift and direct support of the project. The preclinical development presented in this article was funded through grants from Carlsberg Foundation (Sapere Aude grant), Gudbjørg og Ejnar Honorés Fond, Independent Research Fund Denmark (No 0214-00001B (SRP)), a private donation from Line and Mads Brandt Pedersen, and the European Union's Horizon 2020 research and innovation program (No 101003608).

## Author contributions

All authors contributed to: analyzing and discussing the data and proof-reading the manuscript. C.F. and L.G.: writing of the article, CLP, antigen and vaccine design, purification and quality control of the vaccine, mouse studies (ELISA, immunization), planning and designing CLP related experiments. All authors from Aarhus University (M.I., S.R.P., and L.S.R.): designed, performed and analyzed neutralization data. All authors from Expression Biotechnologies (V.S., M.S. (Max Søgaard), J.D., S.C. (Stine Clemensen), B.H., T.D., B.W.N., A.S., M.S (Magdalena Skrzypczak), L.F.A.): design, production, purification, and characterization of protein constructs. All authors from Leiden University (S.K.M., T.J.D., M.K.): designed, performed, and analyzed neutralization data. R.D. and E.W.H.: designing, performing, and analyzing ACE2 binding studies. C.M.J.: designing and performing electron microscopy, DLS measurements, CLP production and purification, CLP, antigen, and vaccine design. K.L.A.: performing ELISA, CLP production and purification, CLP, antigen, and vaccine design. S.M.E., T.G., S.C. (Swati Choudhary), and E.E.V.C.: Large scale development of the VLPs, QC analytical method development for the VLPs. LF: CLP production and purification, CLP, antigen, and vaccine design. ST: contributed to the design of the Tag/Catcher system and CLP design. P.K. and T.M.H.: production, purification, and quality control of monoclonal antibodies used for ELISA studies. M.T., S.K.S., and A.G.S.: antigen design, production, and purification. All authors from Wageningen (L.V.O., G.P.): antigen design, production, and purification. B.M. (Tübingen): application for funding, providing clinical expertise. S.B., A.S.B., J.P.C.: providing support and material for T cell experiments. L.H.H., H.U., K.I.: provided human serum samples, and analysis of it. W.A.J.: creating the COVID consortium, application for funding, design of experiments, supervision of the project. T.G.T., M.A.N., A.S.: application for funding, design of experiments, supervision of the project AFS: supervising the project and writing the article, application for funding, design of experiments.

## Competing interests

C.M.J., S.T., T.G.T., A.S., M.A.N., and A.F.S. are listed as co-inventors on a patent application covering the AP205 CLP vaccine platform technology (WO2016112921 A1) licensed to AdaptVac. Employees of AdaptVac (C.F., L.G., A.F.S., W.A.J.), a company commercializing virus-like particle display technology and vaccine, including several patents. ExpreS$^2$ion employees, as ExpreS$^2$ion is a listed company with IP on ExpreS$^2$ cells. W.D.J. is co-founder and owns ExpreS$^2$ion shares. The other authors have no financial conflicts of interest.

## Additional information

Cyrielle Fougeroux [1,15], Louise Goksøyr [1,2,15], Manja Idorn [3], Vladislav Soroka [4], Sebenzile K. Myeni [5], Robert Dagil [2,6], Christoph M. Janitzek [2], Max Søgaard [4], Kara-Lee Aves [2], Emma W. Horsted [2],

Sayit Mahmut Erdoğan [2,7], Tobias Gustavsson[2,6], Jerzy Dorosz[4], Stine Clemmensen[4], Laurits Fredsgaard [2], Susan Thrane[1], Elena E. Vidal-Calvo [6], Paul Khalifé [2], Thomas M. Hulen [2], Swati Choudhary[2,6], Michael Theisen[2,8], Susheel K. Singh[2,8], Asier Garcia-Senosiain[2,8], Linda Van Oosten[9], Gorben Pijlman [9], Bettina Hierzberger [4], Tanja Domeyer [4], Blanka W. Nalewajek[4], Anette Strøbæk [4], Magdalena Skrzypczak [4], Laura F. Andersson[4], Søren Buus [10], Anette Stryhn Buus[10], Jan Pravsgaard Christensen [10], Tim J. Dalebout[5], Kasper Iversen[11], Lene H. Harritshøj[12], Benjamin Mordmüller [13,14], Henrik Ullum[11], Line S. Reinert [3], Willem Adriaan de Jongh[1,4], Marjolein Kikkert [5], Søren R. Paludan [3], Thor G. Theander[2], Morten A. Nielsen [2✉], Ali Salanti[2,6] & Adam F. Sander [1,2✉]

[1]AdaptVac Aps, 2970 Hørsholm, Denmark. [2]Centre for Medical Parasitology at Department for Immunology and Microbiology, Faculty of Health and Medical Sciences, University of Copenhagen and Department of Infectious Disease, Copenhagen University Hospital, 2200 Copenhagen, Denmark. [3]Department of Biomedicine, Aarhus University, 8000 Aarhus, Denmark. [4]ExpreS2ion Biotechnologies Aps, 2970 Hørsholm, Denmark. [5]Department of Medical Microbiology, Leiden University Medical Center, ZA Leiden 2333, Netherlands. [6]VAR2pharmaceuticals, 2200 Copenhagen, Denmark. [7]Turkish Ministry of Agriculture and Forestry, 06800 Ankara, Turkey. [8]Department for Congenital Disorders, Statens Serum Institute, 2300 Copenhagen, Denmark. [9]Department of Plant Sciences, Laboratory of Virology, 6700AA Wageningen, Netherlands. [10]Department of Immunology and Microbiology, Faculty of Health and Medical Sciences, University of Copenhagen, 2200 Copenhagen, Danmark. [11]Department of Cardiology, Herlev Hospital, 2730 Herlev, Denmark. [12]Department of Clinical Immunology, Copenhagen University Hospital, 2100 Copenhagen, Denmark. [13]Universitätsklinikum Tübingen, Institut für Tropenmedizin, 72074 Tübingen, Germany. [14]Centre de Recherches Médicales de Lambaréné, BP 242 Lambaréné, Gabon. [15]These authors jointly supervised this work: Cyrielle Fougeroux, Louise Goksøyr. ✉email: mortenn@sund.ku.dk; asander@sund.ku.dk

