## [Peer Review File · Nature Communications]

Reviewers' Comments:

Reviewer #1:

Remarks to the Author:

The manuscript by Fougereux et. al. described a rapid SARS-CoV-2 vaccine approach based on a previous nano vaccine platform (AP205 CLPs) expressing SARS-Cov-2 S protein RBD. The study was well designed, data is convincing, and results are interesting to the field. However, the reviewer has identified some concerns to be addressed in a revised version of the submission:

1. Although RBD is a polymeric expression in AP205 CLP, the RBD expression doesn't resemble the state of RBD in the native trimeric S protein. Has any antibody identified recognizing conformational epitopes on RBD? Compared to different nano antigen-displaying platforms for trimeric antigens (like ferritin nanoparticles: Yassine, H. M. et al. *Nat. Med.* 21, 1065–1070; 2015; and Kanekiyo et. al., *Nature*, 2013 Jul 4;499:102-6), what is the advantage of the AP205-CLP for RBD?

2. In the ACE2 binding kinetics studies (Figure 3), positive control of a recombinant S protein may be added to discern if mono RBD is different from trimeric RBD.

3. The authors should discuss if immune responses specific to the nanocarrier can be induced, and if the carrier-specific immunity has any influence on the possible subsequent boost vaccination, given the known short duration of antibodies in recovered patients? The authors may want to discuss how to induce antibody responses with a long duration.

4. Usually, T cell responses last longer and better memory. Any RBD specific T cell responses could contribute to immune protection?

5. Inflammatory cytokine storm is a critical factor for the pathogenesis of SARS-Cov-2. Many inflammatory cytokines are produced in an antibody-dependent manner. The authors may want to discuss if the antibody responses in the studies could worsen the inflammatory responses in the patient lungs in a virus encounter?

Reviewer #2:

Remarks to the Author:

Although more than twenty SARS2 vaccine are under clinical evaluation, the continued development of second-generation vaccines should remain a high priority, even if one or more of these candidates is eventually licensed. Given our current understanding of basic vaccinology in humans, it is unlikely that any of these candidates will be able to induce long term protection, almost certainly not after a single dose. Based on findings with the HPV vaccines, it is likely that a virus-like display vaccine is the only type of non-replicating vaccine that has the potential for meeting this stringent, yet highly desirable, goal. The catcher/tag technology pioneered by this group is, in my opinion, the most attractive strategy for high density, unidirectional, and covalent display of complex targets antigens on the surface of a VLP, and thereby meet the criteria for effective virus-like display. The immunogenicity results reported herein strongly support this opinion. The author's lead candidate generated 100% in vitro neutralizing titers of great than 10,000, a remarkable finding. Importantly, these studies have led to the decision to further evaluate this vaccine in human trials.

Although the overall findings make this one of the more attractive SARS2 vaccine candidate, the study does have limitation. Most importantly, the antibody response to the lead candidate was evaluated using an unusual vaccination protocol (6.5ug prime/<0.1ug boost/ 6.5 ug boost). It seems important to evaluate this vaccine using a more standard prime/boost protocol that would more closely reflect a protocol commonly used in clinical trials. In addition, the dose used in the prime and second boost is on the higher end for a VLP vaccine in mice, and it would have been interest to see results using a lower dose because it might better inform the dosages to evaluate in phase 1 human trials. Further, T cell responses to the vaccine were not evaluated. While it remains debatable whether they will be

important effector mechanisms for effective SARS2 vaccines, VLP-based vaccines tend to induce strong CD4+ and CD8+ T cell responses, and a more complete study would have included these analyses. It is stated that other vaccine candidates using this platform induce a Th1-dominated response in mice, but this predisposition should be verified for this vaccine, at least by assessing vaccine-specific IgG isotype distribution by ELISA in the vaccinated animals.

In terms of potential for large scale industrial production, the AP205 platform is very well suited, since large quantities can be efficiently and inexpensively produced in *E. coli*. However, the catcher/RBD fusion protein was made by transfection of cultured mammalian cells, which is ill suited for this purpose. It is stated that 50mg/L can be generated in a stable cell line. However, if a reasonable estimate of 50 ug per dose in humans is assumed (based on other VLP-based vaccines and the data herein), it seems that an alternative production system will be needed to generate the 100s of million doses needed for a pandemic prophylactic vaccine.

Specific Comments:

1. Line 35: I would advise against coining a new phrase "capsid-like particle" or "CLP" for these vaccines. The definition given in the Intro (line 68) exactly matches that of "virus-like particle" or "VLP" which is the term widely used to describe this type of subunit vaccine in scientific and lay literature. It can only lead to confusion, in that some reader will assume that there must be something fundamentally different between the two types of vaccines.
2. Line 37: please indicate that AP205 is a bacteriophage.
3. Line 110: were higher ratios of RBD to VLP tested to try to increase the density? Based on the structural analyses, does steric hindrance appear to be the limiting factor?
4. Fig 1D: is the RBDn or RBDc construct depicted?
5. Line 166: what is the rationale for choosing Addavax as the adjuvant? For example, if the encapsidate bacterial RNA function as a TLR7/8 agonist, why is an additional adjuvant needed? Is it known to increase antibody response for other AP205-based vaccines?
6. Please report the GMT titers for the ELISA results presented in Fig 4.
7. In Fig 5C, there was 100% neutralization at the highest dilution tested. The dilution series needs to be extended to better understand the remarkable inhibitor activity of these sera.
8. It's unclear whether the Dutch or Danish lab's data is reported in Fig. 5. Both should be reported. There is the same issue with Fig 6.
9. Line 249: this is an example where the terminology may lead to confusion. I have never seen the HPV vaccine called a "CLP".
10. Lines 290-93: the extreme superlatives "ideal platform", "perfect balance" and "complete safety" are impossible to verify and so are inappropriate in this context.

Reviewer #1 (Remarks to the Author):

- 1. Although RBD is a polymeric expression in AP205 CLP, the RBD expression doesn't resemble the state of RBD in the native trimeric S protein. Has any antibody identified recognizing conformational epitopes on RBD?**

We have not yet identified such antibody. However, we are in the process of generating monoclonal antibodies (mAbs) against the RBD. This approach will use the full-length trimeric spike protein for immunization and can thus potentially generate mAbs recognizing conformational epitopes, even those located at the intersection between protomers. Obviously, conformational mAb are great tools for validating the fold of the RBD antigen, however, we believe the binding kinetics studies in the manuscript verify that the rRBD antigens used in this study have a native fold, at least around the ACE2 binding epitope.

- 2. Compared to different nano antigen-displaying platforms for trimeric antigens (like ferritin nanoparticles: Yassine, H. M. et al. Nat. Med. 21, 1065–1070; 2015; and Kanekiyo et. al., Nature, 2013 Jul 4;499:102-6), what is the advantage of the AP205-CLP for RBD?**

We have previously shown that the Tag/Catcher-AP205-CLP platform is well suited for display of trimeric protein antigens (doi: 10.3390/vaccines8030389.). However, we believe our current strategy of focusing the immune response towards only the RBD (incl. the ACE2 binding epitope) (i.e. in contrast to strategies using the full-length spike) is not, as such, dependent on a trimeric antigen presentation, as the ACE2 binding does not seem to be reduced. That said, there is an immediate advantage of the AP205-CLP display platform in that it is a modular system, allowing the antigen to be separately expressed, purified and quality checked (e.g. by measuring Ace2 binding). Moreover, by this 'plug-and-play' approach new antigen variants can quickly be adopted into new vaccines designs *i.e.* in case of mutations arising in the RBD, which lower the efficacy of available vaccines. We have added the following line *"Importantly, the modular Tag/Catcher-AP205 CLP vaccine design makes it possible to replace the current vaccine antigen relatively quickly in the event that the SARS-CoV-2 virus should acquire mutations in the RBD domain reducing the efficacy of an existing vaccine."* in the introduction of the manuscript (lines 70-72).

Finally, aside from these practical aspects, we also believe that the AP205-CLP, in theory, can provide a more optimal epitope display (i.e. due to a closer resemblance to a native virus capsid in terms of epitope distance, size and valency) - leading to increased immunogenicity, although this assertion would have to be experimentally validated.

3. In the ACE2 binding kinetics studies (Figure 3), positive control of a recombinant S protein may be added to discern if mono RBD is different from trimeric RBD.

We appreciate your suggestion and have included ACE2 binding data from a recombinant Full-length trimeric spike protein in supplementary figure 2b.

Although the K_d could not be assessed i.e. due to only having data from a single concentration of FL spike protein. These data show comparable binding kinetics, as was shown for the RBD antigens and CLP-RBD vaccines, at similar concentration. Thus, we do not have any reason to believe that the ACE2 binding affinity is significantly altered in the monomeric RBD antigen compared to the full-length trimeric spike protein. Accordingly, there is also no indication that important neutralizing epitopes located in/around the ACE2 binding region is lost in our rRBD antigens.

We have added the sentence *“Similar binding kinetics were observed for both RBDc (K_D =34.6nM) and full-length spike ectodomain (Supplementary fig. 2A and 2B). This demonstrates that the native structure around the ACE2 binding epitope in the full-length spike protein is maintained when RBDn and RBDc are expressed as soluble proteins.”* (line 155-158)

4. The authors should discuss if immune responses specific to the nanocarrier can be induced, and if the carrier-specific immunity has any influence on the possible subsequent boost vaccination, given the known short duration of antibodies in recovered patients? The authors may want to discuss how to induce antibody responses with a long duration.

We would like to split our answer in two, first addressing the anti-carrier response and secondly, the ability induction of long-lived antibody responses in patients.

The potential issue of pre-existing immunity to VLPs used as carriers in vaccine design (termed carrier induced epitopic suppression (CIES)), has been addressed by Jegerlehner et al (DOI: 10.1016/j.vaccine.2010.02.103). In this study, they investigate if pre-existing immunity against vaccine carrier proteins (VLPs) inhibit the immune response against antigens conjugated to the carrier. The overall conclusions from this study are that CIES depend on the size and coupling density of the conjugated antigen (i.e. how effectively the VLP surface is masked by the antigen), and that CIES pose only a limited negative effect, which can be overcome by higher coupling densities, repeated injections and/or higher doses of the vaccine. We have ourselves, in another study, measured on antigen-specific Ab responses obtained by immunization of mice with/without anti-carrier immunity using the modular Tag/Catcher-AP205 platform and have similarly observed a mild negative CIES effect (submitted MS).

Concerning the specific RBD-CLP vaccine, steric hindrance seem to prevent antigens to bind more than 40-50% of the available 180 binding sites on the VLP surface. This indicates that the CLP surface is very densely decorated by the RBD antigen, suggesting that CIES should be limited. In accordance, we observed a significant boost in Ab titers upon repeated immunizations. We thus also believe that a potential negative effect of CIES could be compensated for by an increased dose.

To answer that question in the manuscript we have added the following lines *“Additionally, the high-density coupling of the RBD antigen to the surface of AP205 CLP is expected to limit anti-CLP immunity from potentially posing a negative effect on the induction of antigen-specific antibody responses upon repeated booster vaccinations”* line 319-321.

Finally, it is possible that our vaccine has potential to induce strong and long-lasting Ab responses in humans (i.e. similar to the HPV VLP vaccine) in which case protection may be achieved after a single dose.

Regarding the longevity of the vaccine-induced immune response, we would like to refer to the review by Schiller & Lowy (DOI: 10.1016/j.vaccine.2017.12.079) explaining the high potency of the HPV vaccines. The remarkable potency of this vaccine may largely be attributed to the structural features of the HPV L1 VLP, which lead to the efficient generation of long-lived antibody-producing plasma cells. Due to the great similarity in architecture and epitope display between HPV L1 CLP and our RBD-AP205 CLP, respectively, it can be assumed that the two vaccines may induce a similar type of immune response. Thus, we expect that the RBD-CLP vaccine holds potential to induce a strong and long-lived antigen-specific immune response, as seen for the HPV vaccine.

5. Usually, T cell responses last longer and better memory. Any RBD specific T cell responses could contribute to immune protection?

It is still not known to what extent T-cell responses is important for COVID-19 vaccine efficacy, and it is clear that our VLP vaccine mainly focus on the antibody responses. We have now investigated the IgG subclass profile and measured CD4+ T-cell responses (respectively figure 4 and supplementary figure 4). Compared to immunization with monomeric RBD, the RBD-CLP vaccine induces relatively more IgG2a/2b compared to IgG1 *i.e.* indicating induction of a stronger TH1-type immune response. The following lines were added about the IgG subclass profile:

“Finally, the IgG subclass profile was assessed for the antibody responses induced by the soluble RBDn and RBDn-CLP vaccine, respectively. The soluble RBDn vaccine induces a Th2-biased immune response

indicated by a predominant induction of IgG1 antibodies (Fig. 4B), while the corresponding RBD-CLP vaccine induce a more balanced Th1/Th2-type response with significantly increased levels of IgG2a and IgG2b. Similar results were shown for the RBDc-CLP vaccine (data not shown)." Line 179-183

"Additionally, our data show that delivery of the RBD antigen by the Tag/Catcher-AP205 platform promotes induction of IgG2a and IgG2b subclasses (i.e. characteristic of a Th1 response), which is believed to reduce the potential risk of vaccine-related enhancement of disease⁶³." Line 291-293

Additionally, we have measured induction of CD4+ T cells by the VLP vaccine and see that the vaccine induce a minor but statistically significant CD4+ response, supporting the TH1-type immune response.

Regarding the T cell responses, the line *"This was further supported by FACS analysis showing statistically increased levels of RBD-specific CD4+ IFN γ + T cells in mice immunized with the RBDn-CLP vaccine (Supplementary fig. 4)."* was added line 183-185.

- 6. Inflammatory cytokine storm is a critical factor for the pathogenesis of SARS-CoV-2. Many inflammatory cytokines are produced in an antibody-dependent manner. The authors may want to discuss if the antibody responses in the studies could worsen the inflammatory responses in the patient lungs in a virus encounter?**

We are already discussing the fact that our strategy of focusing the immune responses towards neutralizing epitopes in the RBD may result in induction of a higher relative proportion of neutralizing over binding antibodies *i.e.* compared to vaccines delivering the full-length spike protein. Additionally, we have now added new data and elaborated on the fact that our vaccine support a Th1-type immune response, which is believed to minimize the risk of vaccine-associated enhanced disease.

Reviewer #2 (Remarks to the Author):

- 7. Most importantly, the antibody response to the lead candidate was evaluated using an unusual vaccination protocol (6.5ug prime/<0.1ug boost/ 6.5 ug boost). It seems important to evaluate this vaccine using a more standard prime/boost protocol that would more closely reflect a protocol commonly used in clinical trials.**

We fully understand your comment, and agree that the immunization regimen is unusual. This was the first data produced for our lead candidate vaccine. However, later mouse studies testing the RBDn-CLP vaccine were performed in a classical prime-boost regimen (5 μ g per dose), which showed comparable antibody kinetics. The sentence "Antibody titers measured in sera from BALB/c mice immunized with a

more standard prime-boost regimen (5 μ g of RBDn-CLP), were not significantly different from the titers shown in Fig. 4 (Supplementary fig. 3).” has been added to the manuscript (line 177-179).

- 8. In addition, the dose used in the prime and second boost is on the higher end for a VLP vaccine in mice, and it would have been interest to see results using a lower dose because it might better inform the dosages to evaluate in phase 1 human trials.**

For the RBDc-CLP vaccine we show data in the manuscript from mice immunized in a prime-boost with 1 μ g. This data both show lower ELISA binding titers, as well as lower neutralization capacity.

- 9. Further, T cell responses to the vaccine were not evaluated. While it remains debatable whether they will be important effector mechanisms for effective SARS2 vaccines, VLP-based vaccines tend to induce strong CD4+ and CD8+ T cell responses, and a more complete study would have included these**

analyses. It is stated that other vaccine candidates using this platform induce a Th1-dominated response in mice, but this predisposition should be verified for this vaccine, at least by assessing vaccine-specific IgG isotype distribution by ELISA in the vaccinated animals.

We agree this should be verified for the RBD-CLP vaccine. We have now investigated the IgG subclass profiles in sera obtained from mice immunized with the monomeric RBD and RBD-CLP vaccines, respectively. This data confirm that the CLP vaccine, in fact, induces more IgG2a/2b, indicating a stronger TH1-type immune response. Additionally, we have measured induction of CD4+ T cells by the nRBD-CLP vaccine and see that the vaccine induce a minor but statistically significant CD4+ IFN γ + response, supporting the TH1-type immune response. Most other studies use Elispot as a detection method and longer incubation with peptide, which both can increase the sensitivity. However, the IgG subclass and T-cell data have now been included in the manuscript in figure 4 E-F and supplementary figure 4, respectively (incl. line 179-185 in the MS).

- 10. In terms of potential for large scale industrial production, the AP205 platform is very well suited, since large quantities can be efficiently and inexpensively produced in E. coli. However, the catcher/RBD fusion protein was made by transfection of cultured mammalian cells, which is ill suited for this purpose. It is stated that 50mg/L can be generate in a stable cell line. However, if a reasonable estimate of 50 ug per dose in humans is assumed (based other VLP-based vaccines and the data herein), it seems that an alternative production system will be needed to generate the 100s of million doses need for a pandemic prophylactic vaccine.**

We agree that the current yield of 50mg per liter may be a problem for large-scale production. However, this is based on using a polyclonal virus stock. After selection of a specific clone, the yield can be expected to increase by more than 10 fold.

Specific Comments:

- 11. 1. Line 35: I would advise against coining a new phrase “capsid-like particle” or “CLP” for these vaccines. The definition given in the Intro (line 68) exactly matches that of “virus-like particle” or “VLP” which is the term widely used to describe this type of subunit vaccine in scientific and lay literature. It can only lead to confusion, in that some reader will assume that there must be something fundamentally different between the two types of vaccines.**

We have recently introduced the CLP phrase in a couple of already published articles, with the purpose to distinguish capsid-based VLPs from lipid-based VLPs (e.g. baculovirus, HBV, Novavax etc.) since we believe these are often evaluated as a common technology, and we believe there are several practical and immunological differences between these types of VLPs. We would thus like to keep this explanation in this paper as well, so that it is in accordance with our previous publications.

- 12. Line 37: please indicate that AP205 is a bacteriophage.**

This has been clarified in line 69.

- 13. Line 110: were higher ratios of RBD to VLP tested to try to increase the density? Based on the structural analyses, does steric hindrance appear to be the limiting factor?**

The following elaborating sentence has been added to the manuscript:

“A higher coupling efficiency could not be obtained by increasing the molar excess of antigen, indicating steric hindrance.” (line 117-118)

- 14. Fig 1D: is the RBDn or RBDc construct depicted?**

In Fig. 1D a theoretical model, which is applicable to both the nRBD and cRBD antigens coupled to the AP205 platform is shown. Theoretically, the orientation of the RBDc and RBDn would not differ significantly, as the N- and C-termini of recombinant RBD is located very close to each other, as explained in the results section (line 121-123).

15. Line 166: what is the rationale for choosing Addavax as the adjuvant? For example, if the encapsidate bacterial RNA function as a TLR7/8 agonist, why is an additional adjuvant needed? Is it known to increase antibody response for other AP205-based vaccines?

We currently have a paper in review in which we test a panel of clinically relevant extrinsic adjuvants in combination with the Tag/Catcher AP205 CLP platform. In this study, Addavax has performed quite well in combination with the CLP platform. Additionally, the choice was made upon what was actually available for us to use in our future clinical trial.

16. Please report the GMT titers for the ELISA results presented in Fig 4.

Thank you for your comment. GMT has been added to the figure legend of figure 4 for each measurement.

17. In Fig 5C, there was 100% neutralization at the highest dilution tested. The dilution series needs to be extended to better understand the remarkable inhibitor activity of these sera.

Thank you for your comment. We have now performed a further dilution of the sera and included this in the figure 5C as “additional dilution experiment”.

18. It's unclear whether the Dutch or Danish lab's data is reported in Fig. 5. Both should be reported. There is the same issue with Fig 6.

This has now been clarified in figure 5-6 and supplementary figure 5-7 legends. However, both the Danish and Dutch lab's data are depicted in the article and in supplementary figures, respectively.

19. Line 249: this is an example where the terminology may lead to confusion. I have never seen the HPV vaccine called a “CLP”.

We agree with your comment. This was a misspelling and has been corrected to “VLP”.

20. Lines 290-93: the extreme superlatives “ideal platform”, “perfect balance” and “complete safety” are impossible to verify and so are inappropriate in this context.

We agree, and have now edited these sentences accordingly.

Reviewers' Comments:

Reviewer #1:

None

Reviewer #2:

Remarks to the Author:

I'm satisfied with the replies of the authors and revision made in the manuscript in response to the issues raised in my review. I still don't like the new term "CLP" but I grant the author's discretion to use the term they prefer.